# Characteristics of Pica Behavior among Mothers around Lake Victoria, Kenya: A Cross-Sectional Study

**DOI:** 10.3390/ijerph16142510

**Published:** 2019-07-14

**Authors:** Esther O. Chung, Brian Mattah, Matthew D. Hickey, Charles R. Salmen, Erin M. Milner, Elizabeth A. Bukusi, Justin S. Brashares, Sera L. Young, Lia C.H. Fernald, Kathryn J. Fiorella

**Affiliations:** 1Department of Epidemiology, Gillings School of Global Public Health, University of North Carolina at Chapel Hill, McGavran-Greenberg Hall, CB #7435, Chapel Hill, NC 27599-7435, USA; 2Ekialo Kiona Research Dept, Organic Health Response, Mbita, P.O. Box 224-40305, Kenya; 3Division of HIV, Infectious Disease, and Global Medicine, Department of Medicine, UCSF, 1001 Potrero Ave, San Francisco, CA 94110, USA; 4Department of Family Medicine and Community Health, University of Minnesota, 5516 Delaware St SE, Minneapolis, MN 55455, USA; 5Bureau for Global Health, USAID, Washington, DC 20001, USA; 6Centre for Microbiology Research, Kenya Medical Research Institute, Hospital Road, Nairobi 00100, Kenya; 7Department of Environmental Science, Policy, and Management, University of California, Berkeley, 130 Mulford Hall #3114, Berkeley, CA 94720, USA; 8Department of Anthropology, Northwestern University, 1810 Hinman Avenue, Evanston, IL 60208, USA; 9Division of Community Health Sciences, School of Public Health, University of California, Berkeley, 2121 Berkeley Way, Room 5302, Berkeley, CA 94720-7360, USA; 10Master of Public Health Program, Department of Population Medicine and Diagnostic Sciences, Cornell University, S2-004 Shurman Hall, Ithaca, NY 14853, USA

**Keywords:** pica, geophagy, amylophagy, pregnancy, breastfeeding, Kenya

## Abstract

*Background*: Pica, the craving and purposeful consumption of nonfoods, is poorly understood. We described the prevalence of pica among women on Mfangano Island, Kenya, and examined sociodemographic and health correlates. *Methods*: Our cross-sectional study included 299 pregnant or postpartum women in 2012. We used a 24-h recall to assess pica, defined as consumption of earth (geophagy), charcoal/ash, or raw starches (amylophagy) and built multivariable logistic regression models to examine sociodemographic and health correlates of pica. *Results*: Eighty-one women (27.1%) engaged in pica in the previous 24 h, with 59.3% reporting amylophagy and 56.8% reporting geophagy, charcoal, and/or ash consumption. The most common substances consumed were raw cassava (*n* = 30, 36.6%), odowa, a chalky, soft rock-like earth (*n* = 21, 25.6%), and soil (*n* = 17, 20.7%). Geophagy, charcoal, and/or ash consumption was negatively associated with breastfeeding (OR = 0.38, 95% CI: 0.18–0.81), and amylophagy was associated with pregnancy (OR = 4.31, 95% CI: 1.24–14.96). Pica was more common within one of six study regions (OR = 3.64, 95% CI: 1.39–9.51). We found no evidence of an association between food insecurity and pica. *Conclusion*: Pica was a common behavior among women, and the prevalence underscores the need to uncover its dietary, environmental, and cultural etiologies.

## 1. Introduction

Pica is the intense craving and purposeful consumption of substances not commonly identified as food, and pica has been documented in numerous cultures [1,2,3]. While the types of substances consumed range widely, the most frequent forms of pica are geophagy (consumption of earth) and amylophagy (consumption of raw starches) [3]. Pica is common among vulnerable populations, especially pregnant women and children, and has been associated with both positive and negative health effects [3,4,5]. Some evidence has shown that pica can relieve gastrointestinal distress [6,7,8] and protect against harmful pathogens [9,10,11]. Yet, pica has also been associated with heavy metal poisoning [12], iron deficiency [13,14,15,16,17], and helminth infections [16,18].

The etiology of pica is not well understood, but various hypotheses have been proposed. The hunger hypothesis suggests that pica is motivated by hunger and that individuals engage in pica to replace nutrients not found in their diet [19,20]. Specifically, geophagic substances, such as clay and earth, are consumed in order to satiate hunger [3]; however, this hypothesis cannot explain all pica behavior given that individuals commonly report their motivation as intense cravings, not hunger [2,20]. The nutritional hypothesis suggests pica is a biological response to micronutrient deficiencies, namely iron, calcium, or zinc [2,21], and the protective hypothesis suggests pica relieves short-term illnesses and/or long-term effects of chemicals, pathogens, and parasites [2]. Studies have identified certain pica substances, such as clays, that can bind directly to pathogens or toxins [22,23,24] or reduce the intestinal wall’s permeability, which could prevent transmission of harmful pathogens [25,26]. Finally, the cultural hypothesis suggests that factors such as cultural beliefs or social norms may contribute to pica behavior [19,20]. Given that cultural and societal norms indubitably shape human behaviors, it is likely that such norms reinforce or inhibit pica. Previous research among coastal Luo Kenyans found that geophagy during pregnancy was attributed to a cultural association with blood and fertility [27]. Moreover, food insecurity and socioeconomic status have been posited to influence pica, and likely contribute to the hunger and nutritional hypotheses [2].

Previous studies in Kenya found a high prevalence of pica among pregnant and postpartum women in urban and rural settings [27,28,29,30,31,32,33]. In a coastal region of Kenya, 73% of pregnant women reported consumption of soil regularly during their pregnancy [27]. Another study in coastal Kenya found that 56% of pregnant women reported a history of eating soil regularly, and those who participated in geophagy were more likely to be anemic [28]. Studies in the Bondo District of Western Kenya found a high prevalence of geophagy among pregnant mothers and that 30% still consumed earth or soil at six months postpartum [29,30,31]. In these Kenyan studies, pregnant and lactating women were interviewed about their history of pica behavior, namely geophagy. While a longer recall period potentially provides valuable information about rare behaviors, it is vulnerable to recall bias, as most individuals had participated in pica at some point, such that disaggregating proximate factors becomes more difficult. A shorter recall period can help to pinpoint more temporally adjacent factors associated with pica.

Therefore, we conducted the present study on Mfangano Island, a 65 km^2^ continental island in Lake Victoria, Kenya, with a population of approximately 25,000 people, mostly of Luo and Suba descent. The island is situated in Homa Bay County, Kenya, where the estimated HIV prevalence among adults is approximately 26% [34]. The introduction of non-native Nile perch in the 1950s caused drastic declines in the biodiversity of the lake [35] and spurred an export fishing industry. Recent research demonstrated declines in fish catch [36,37], and regional communities continue to experience high levels of food insecurity and poverty despite proximity to fish resources [38]. Residents on the island typically rely on fishing and subsistence farming for food and income; however, research has shown that fishers do not eat more fish than their nonfishing neighbors [39].

Mfangano Island contains four regions along the lakeshore, and we disaggregated regions further to also include a community that lives in the center of the island away from the lakeshore and one on a nearby satellite island. We omit regional names due to the potential stigma surrounding pica and consequences of associating these communities with pica behavior. All regions experience similar levels of food insecurity, with local residents primarily reliant on the Lake Victoria fisheries for Nile perch and other fish for their food and income. The satellite island, like many sites on Mfangano, has a commercially active fish landing site though differs in its agricultural development, with a rocky soil environment that reduces participation in maize and vegetable farming relative to sites on Mfangano.

Our objectives were to: (1) quantify the prevalence of pica among mothers with young children on Mfangano Island using a short recall period, and (2) explore the proximate sociodemographic and health correlates of pica. Given the background literature and context, we predicted that pregnancy and breastfeeding, maternal morbidity, and food insecurity would be associated with pica behavior.

## 2. Methods

### 2.1. Study Population

This cross-sectional study used baseline data collected between December 2012–March 2013 from a larger prospective study examining fishing livelihoods, fish consumption, and early child nutrition [39,40]. Details on the larger study have been described elsewhere [41,42]. Briefly, households (*N* = 303) were selected using stratified random sampling proportional to regional population in which Regions 1–5 were on the main island and Region 6 describes the satellite island: Region 1 = 66; Region 2 = 16; Region 3 = 81; Region 4 = 46; Region 5 = 60; and Region 6 = 34. Given the larger study’s focus on early child nutrition, enrollment criteria were (1) having at least one child less than 2 years of age and (2) living on Mfangano Island. Data on all covariates were available for 299 households.

Local enumerators conducted surveys in the local language, Dholuo, and data collection tools were developed from validated measures and locally adapted. Heads of households provided consent to participate in the study, with women providing consent for their own participation. The Committee for Protection of Human Subjects at the University of California, Berkeley and the Ethical Review Committee at the Kenya Medical Research Institute approved the study protocol (CPHS 2010-01-608; SSC 2334).

### 2.2. Study Variables

#### 2.2.1. Outcome

Women self-reported pica during a 24-h recall survey in which they were asked if they had consumed any of the following: ash, charcoal, odowa (a chalky, soft rock-like earth), soil/other, uncooked foods (cassava, rice, etc.), and other nonfood items (see Appendix A). We operationalized pica into three categories: (1) the consumption of earth, soil, or clay (geophagy) and/or charcoal/ash, (2) amylophagy, the consumption of raw starchy foods, and (3) any nonfood craving. We combined geophagy and charcoal/ash consumption due to the small number of women who consumed charcoal (*n* = 9, 11.1%) or ash (*n* = 4, 4.9%), and their physical properties being similar to those of earth.

#### 2.2.2. Key Covariates

Sociodemographic information was collected using standardized questionnaires. Maternal education was quantified as the highest level attained and categorized into none or some primary school, primary school, some secondary school, secondary school or higher. We used principal components analysis to create an asset index that included ownership of items such as electricity, livestock, and type of flooring or roofing materials [43]; however, as eigenvalues did not account for substantial variation, we retained a total count as our asset score. Household food insecurity was measured with the Household Food Insecurity Access Scale, a 9-item questionnaire that generates a food insecurity score from 0–27 [44]. A maternal morbidity score was calculated using the Medical Outcomes Survey–HIV (MOS–HIV), a validated, self-reported measure of health-related quality-of-life, and was centered at 0 and normalized by standard deviation [45,46]. While the MOS–HIV metric is correlated with measures of HIV disease progression (e.g., CD4 count), we did not collect information about HIV status [47]. During the 24-h recall, women were also asked if they were ill in the prior day, if they consumed any tablets, herbs, or medicine and what types, and if they consumed any deworming medications in the last 3 months. Maternal breastfeeding practices and pregnancy status were ascertained using maternal self-report. We also included the six regions of the study site as indicator variables to assess regional differences.

### 2.3. Statistical Analysis

We first descriptively characterized the sample and the types of pica substances women reported consuming. We then conducted multivariable logistic regression models for any pica, geophagy/charcoal/ash, and amylophagy based on a priori identification of variables that have been hypothesized to be associated with pica behavior in previous studies and included maternal age, maternal education, current pregnant and breastfeeding status, morbidity score, food insecurity, and number of people in the household. We included region in the models because we hypothesized the potential for cultural transmission of pica behavior and environmental differences in access to common pica materials (e.g., odowa). Maternal health factors (e.g., being ill the day before, and consumption of iron tablets, antimalarial or antiretroviral drugs, or deworming medication) were initially hypothesized to be associated with pica; however, these were omitted to improve estimation as they were not significantly different between groups that did and did not participate in pica, and their omission did not alter the magnitude of coefficients of the final models. We also conducted a sensitivity analysis to include interviewer fixed effects and did not find substantial differences in estimates (Appendix A). All statistical analyses were performed using Stata 14 [48].

## 3. Results

Eighty-one (27.1%) women reported engaging in any form of pica in the previous 24 h (Figure 1). Among the 81 women reporting any pica behavior, the prevalence of geophagy, charcoal, and/or ash consumption was 56.8% (*n* = 46) and the prevalence of amylophagy was 59.3% (*n* = 48). Thirteen women reported both geophagy, charcoal, and/or ash consumption and amylophagy. Of the types of pica reported, most women consumed raw cassava (*n* = 30, 36.6%), odowa (*n* = 21, 25.6%), and soil (*n* = 17, 20.7%). Among the 303 participants, on average, women were 28.1 years (SD: 8.4), 90.8% were married, 50.2% had no education or completed some primary school, and 73.6% were breastfeeding (Table 1). Stratified sample characteristics by any pica, geophagy/charcoal/ash, and amylophagy are reported in Table 1. 

In multivariable logistic regression models, we found the odds of engaging in any form of pica in Region 6, the satellite island, were 3.64 times (95% CI: 1.39–9.51) that in Region 1, a site on Mfangano Island, after controlling for maternal age and education, household assets, and number of people in the household (Table 2). Women living in Region 6 had 3.73 times the odds of earth, charcoal, and/or ash consumption compared to women living in Region 1 (95% CI: 1.27–11.00), controlling for all other covariates. However, wide confidence intervals for the regional estimates reflect low precision due to the small sample size of women living in Region 6 (*n* = 36). The odds of consuming geophagy (earth), charcoal, and/or ash were lower among women who were currently breastfeeding (OR = 0.38, 95% CI: 0.18–0.81). Pregnant women had 4.31 times the odds of amylophagy (consumption of raw starches) compared to nonpregnant (95% CI: 1.24–14.96), controlling for all other covariates. There was also low precision due to the small number of women who were currently pregnant and reported amylophagy (*n* = 7).

## 4. Discussion

Our study demonstrates that pica is a common behavior among pregnant and postpartum women on Mfangano Island in Lake Victoria, Kenya, given that more than 25% of women participated in pica in the previous 24 h. Amylophagy was common, and geophagy, charcoal, and/or ash consumption was also widespread. The most commonly reported substances were raw cassava, odowa, and soil. Yet, we also found limited associations between pica behavior and possible explanatory variables.

While previous studies in Kenya reported the prevalence of pica between 45%–73% among pregnant mothers [28,29,30,31,32], our population was mainly composed of nonpregnant, lactating mothers (221 out of 299 mothers). The prevalence of pica among pregnant women, a small subset of our sample, found that 9 out of 17 women (52.9%) consumed nonfood items, which is comparable to estimates from other studies in the same region. Given the small sample size of pregnant women, our confidence interval was wide for the association between pregnancy and amylophagy (95% CI: 1.24–14.96); however, the point estimate (OR = 4.31) suggests there is a strong, positive association, which is corroborated by prior evidence that pregnant women are likely to engage in pica [2,49]. Our findings also indicate that pica is a behavior that is not limited to pregnancy, with 53 out of 221 women (24%) reporting any pica consumption while currently breastfeeding. Moreover, we found that women who were currently breastfeeding had lower odds of participating in the consumption of earth, charcoal, and/or ash. However, we acknowledge that the postpartum period is a distinct transition period whereby women may be returning to work and chores, which may limit their engagement in pica behavior.

We expected to find associations between pica and greater maternal morbidity and more food insecurity; however, we found no evidence for any of the three outcomes. Our morbidity measure was a health-related quality-of-life measure that summarized mental and physical health but did not specifically assess gastrointestinal distress. Our null findings for food insecurity may be due to the shorter, 24-h recall period that we used to define pica, which did not directly match the 30-day recall period of the food insecurity measure. Another explanation includes our small sample size and the homogeneity of the sample, in which 66% (197/299) were categorized as severely food-insecure, which means our findings largely compare gradations of food insecurity, rather than making comparisons to a food-secure group. Future research should consider other measures of maternal health, such as perceived stress, depression, and anxiety, which have been found to be associated with pica [20,50]. Additionally, food insecurity and pica frequency measures with varying recall periods (e.g., 24-h, 7-day, and 30-day) would help to provide insight into how resource scarcity affects pica consumption patterns.

We found pica was approximately twice as common in Region 6, where 50.0% (17/34) of women engaged in pica compared to 21.2%–27.2% in other regions. In this region, consumption of odowa (20.6%) and charcoal or ash (11.8%) was 3.2 and 1.6 times higher, respectively, than in any other region. This particular region also reported the highest rates of amylophagy (24.0%, 8/34), relative to the other regions (12.1%–21.7%). This finding suggests that contextual factors, which may include cultural norms or dietary quality, may play an important role in driving pica behavior. Although many features of Region 6 are similar to those of other regions of Mfangano Island, several unique aspects of this community and its relative isolation may create a different context for pica behavior. The population resides in a relatively densely populated satellite island compared to a population that resides in fish-landing sites and rural villages on the main island. In addition, the terrain of the region is rocky, with limited farming space. Thus, agricultural activities are more constrained in comparison to other regions, perhaps reducing household access to fruits, vegetables, meat, and eggs. Further, previous research supports that consumption of fish is driven by income rather than participation in fishing livelihoods [39]. The physical environment combined with commercialized fishing, patterns of migration, and constrained access to micronutrient-rich foods may contribute to the higher rates of pica observed in this region. Though additional research is necessary to confirm these pathways, this strong geographical finding suggests that pica may be localized and not dispersed across regions, as previously reported [28,29,30,31,32].

While we were unable to assess how cultural beliefs affected pica in our sample, it cannot be ignored as an important contributor. Pica may be a strongly embedded cultural practice among some groups of women in island communities around Lake Victoria, and its social and cultural significance may hold importance for users beyond its physiological effects. Among the Luo population on the Kenyan coast, pregnant women have previously reported that the practice of geophagy was strongly connected to fertility, healthy blood during reproduction, and community [27]. The consumption of soil was reported as a predominately female practice during pregnancy that was typically not discussed with strangers given the potential for ridicule and mockery, particularly by men. While this study was in coastal Kenya, the same cultural beliefs may be present in and around Lake Victoria. This type of stigma in communities may lead to underreporting of pica substances.

Strengths of this study include examining pica for the first time in this lakeside population, measuring pica with a short recall period, and including a previously underexplored population of postpartum women. However, there are some limitations of our study. The cross-sectional nature limits our ability to establish causality, and our findings associated with pregnancy are limited given the study cohort did not specifically target and sample pregnant women. Further, our measure of pica consumption lacked details to rigorously examine etiologic hypotheses. Future research should include more in-depth survey information on pica such as quantities and frequencies of substances consumed, and potential causes and consequences of pica, including micronutrient deficiencies, gastrointestinal malaise, and food insecurity [7]. Additionally, prior research has demonstrated that local definitions of food vs. nonfood affected reporting of geophagic substances [19,20]. Though we did not directly collect women’s perception of whether uncooked foods were considered food or nonfood, informal conversations within the Mfangano Island community found that women who consume uncooked starches (i.e., uncooked cassava or uncooked rice) use them to satisfy cravings rather than as food, which suggests women consider these items as nonfood in the same category as items such as odowa. It is also possible that there are seasonal differences in pica behavior based on food availability, which we were not able to control for, as data were only collected from December 2012 to March 2013. While our study had the potential for recall bias, this was minimized due to the short 24-h recall period; however, stigma related to the consumption of nonfood items may lead to biased reporting and an underestimate of pica prevalence in our population. Moreover, while 24-h recalls are commonly validated tools for dietary intake, we acknowledge that a 24-h recall period may not be representative of typical intake of nonfood items. Future work should include both short and long recall periods in order to understand drivers of pica behavior and changes in pica consumption patterns over different time periods (e.g., during pregnancy and postpartum). Additionally, the homogeneity of the study population could affect associations such that we were not able to detect differences (i.e., most of the women on the island have low education levels (none to primary school), low economic status, and high food insecurity scores). Finally, combining the raw, starchy foods into one category may mask underlying associations given that they may have different macro- and micronutrient contents. Related, combining geophagy, charcoal, and/or ash consumption may also hide different etiologies and health consequences.

Given the commonality and potential benefits and harm of this practice, more research is needed to better understand the drivers and consequences of this behavior among communities in and around Lake Victoria and throughout East Africa. A combination of shorter (i.e., 24-h or 30-day) and longer (i.e., history of pica since childhood) recall periods in addition to longitudinal data with repeated measures of pica will provide more nuanced insight into the causes and consequences. Moreover, research into these pathways could provide further information to test etiologic hypotheses. In order to rigorously examine the causes of pica around Lake Victoria, future studies should use objective biomarkers to measure iron and zinc micronutrient deficiencies, collect soil and odowa composition and intake, and gather information on gastrointestinal illnesses and discomforts. Moreover, studies could include information about dietary intake, such as animal source food frequency to assess micronutrient consumption. To examine the underlying causes of the high prevalence of pica in Region 6, future studies should also collect qualitative data through in-depth interviews and focus groups on the significance of pica and perceptions in the community in order to unpack how culture and social norms influence pica. Additional measures of resource scarcity should expand beyond physical resources, such as food insecurity and assets, and include personal and social resources, such as empowerment, decision making, and community cohesion, which could shed light on differences in pica reporting and consumption patterns.

## 5. Conclusions

Our study demonstrates that pica is a common behavior on Mfangano Island, with one in four mothers with young children engaging in the consumption of nonfood items in the previous 24 h. In particular, we found a higher prevalence of pica on a satellite island where half of all women reported participating in pica. In our data, the observed null relationships between socioeconomic and health covariates and pica underscore the complex nature of pica, which is impacted by numerous potential influences—physical, economic, and cultural. As little is known about the causes of pica and its long-term effects, an improved understanding of this relatively common behavior merits further examination.

## Figures and Tables

**Figure 1 ijerph-16-02510-f001:**
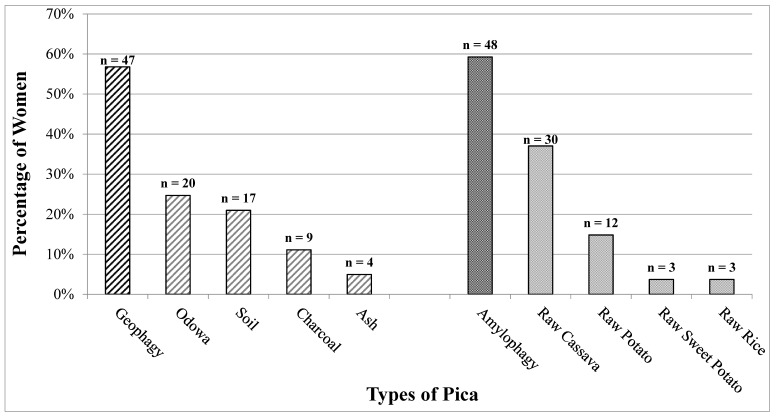
Types of pica substances reportedly consumed among women in the previous 24 h. Percentages were calculated out of the total number of women reporting any form of pica (*n* = 81). Sixteen women reported consuming two or more substances, with *n* = 13 reporting both geophagy, charcoal, and/or ash consumption and amylophagy.

**Table 1 ijerph-16-02510-t001:** Sample characteristics of mothers with children under 2 years, Mfangano Island, Kenya.

Characteristic	Mean ± SD or *N* (%)			
	Whole Sample (*n* = 299)	Any Pica (*n* = 81)	Geophagy/Charcoal/Ash (*n* = 46)	Amylophagy (*n* = 48)
Sociodemographic				
Age (years)	28.2 ± 8.4	26.7 ± 6.5	26.9 ± 6.6	27.3 ± 7.4
Married	271 (90.6%)	71 (87.7%)	42 (91.3%)	41 (85.4%)
Number of People in Household (Range: 2–12)	5.7 ± 1.9	5.6 ± 1.9	5.6 ± 1.8	5.6 ± 2.1
Highest Education Level				
None or Some Primary	150 (50.2%)	43 (53.1%)	24 (52.2%)	28 (58.3%)
Primary	91 (30.4%)	21 (25.9%)	15 (32.6%)	8 (16.7%)
Some Secondary	29 (9.7%)	10 (12.4%)	3 (6.5%)	8 (16.7%)
Secondary or Higher	29 (9.7%)	7 (8.6%)	4 (8.7%)	4 (8.3%)
Household Asset Score (Range: 1–9)	1.9 ± 1.2	1.9 ± 1.1	2.0 ± 1.1	1.7 ± 1.1
Employed (*n* = 222)	186 (62.2%)	47 (58.0%)	25 (54.4%)	30 (62.5%)
Food Insecurity Score (Range: 0–27)	9.3 ± 5.1	9.8 ± 4.8	9.6 ± 4.6	9.8 ± 5.2
Reproductive Status				
Currently Pregnant	17 (5.7%)	9 (11.1%)	5 (10.9%)	7 (14.6%)
Currently Breastfeeding	221 (73.9%)	53 (65.4%)	27 (58.7%)	34 (70.8%)
Health				
Ill Yesterday	51 (17.1%)	14 (17.3%)	6 (13.0%)	9 (18.8%)
Current Medications				
Antimalarial	23 (7.7%)	10 (12.4%)	4 (8.7%)	6 (12.5%)
Antiretroviral	22 (7.4%)	5 (6.2%)	3 (6.5%)	3 (6.3%)
Deworming medication in last 3 months	24 (8.0%)	9 (11.1%)	3 (6.5%)	8 (16.7%)
Morbidity Score	0 ± 1.0	0 ± 1.0	−0.1 ± 0.9	0 ± 1.0
Region				
1	65 (21.7%)	13 (16.1%)	8 (17.4%)	7 (14.6%)
2	16 (5.4%)	4 (4.9%)	3 (6.5%)	2 (4.2%)
3	79 (26.4%)	22 (27.2%)	13 (28.3%)	11 (22.9%)
4	46 (15.4%)	12 (14.8%)	4 (8.7%)	10 (20.8%)
5	59 (19.7%)	13 (16.1%)	6 (13.0%)	10 (20.8%)
6	34 (11.4%)	17 (21.0%)	12 (26.1%)	8 (16.7%)

**Table 2 ijerph-16-02510-t002:** Correlates of any pica, geophagy/charcoal/ash, and amylophagy, Mfangano Island, Kenya (*n* = 299) *.

Characteristic	Any Pica	Geophagy/Charcoal/Ash	Amylophagy
	OR (95% CI)	OR (95% CI)	OR (95% CI)
Currently Pregnant	2.52 (0.77–8.29)	1.25 (0.32–4.84)	4.31 (1.24–14.96)
Currently Breastfeeding	0.54 (0.28–1.04)	0.38 (0.18–0.81)	1.09 (0.47–2.52)
Maternal Morbidity Score	0.78 (0.58–1.07)	0.79 (0.54–1.16)	0.86 (0.59–1.24)
Household Food Insecurity Score	1.06 (0.99–1.13)	1.04 (0.96–1.12)	1.04 (0.96–1.12)
Region			
1	Referent	Referent	Referent
2	1.17 (0.30–4.58)	1.74 (0.37–8.29)	0.95 (0.16–5.49)
3	1.72 (0.75–3.94)	1.47 (0.55–3.95)	1.52 (0.52–4.40)
4	1.40 (0.54–3.62)	0.62 (0.17–2.30)	2.34 (0.76–7.20)
5	1.17 (0.47–2.91)	0.83 (0.26–2.66)	1.66 (0.56–4.95)
6	3.64 (1.39–9.51)	3.73 (1.27–11.00)	2.11 (0.64–6.97)

* Separate multivariable models were run for any pica, geophagy/charcoal/ash, and amylophagy. All models controlled for maternal age, education, household assets, and number of people in the household.

## Data Availability

The data and materials related to this manuscript are available from the corresponding author upon reasonable request.

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
