# Peer review of "Characteristics of Pica Behavior among Mothers around Lake Victoria, Kenya: A Cross-Sectional Study"

_ijerph, 2019, doi:10.3390/ijerph16142510_

Round 1
Reviewer 1 Report
Summary
Thank you for inviting me to be a reviewer on this paper. This is an interesting topic, but the paper needs a lot of revisions in order to be considered for publication.
The goal of this paper was to examine the sociodemographic and health correlates of pica among women in Kenya. At first, I thought this study examining pica behavior among an often overlooked population, non-pregnant and breastfeeding women, however the study actually does include pregnant women which is not mentioned until later. Given that lack of detail, I think the contribution of this study is not clear from the get-go.
Also, the authors discuss the hypotheses for pica but do not say that they are testing them, but then later apply different variables to the hypotheses for pica in the discussion. For example, for the first half of the paper – especially once I read the study population section – I thought the authors could test the hunger/micronutrient hypothesis because of the extensive detail about the population being food insecure, yet there is no inclusion of that hypothesis, or any other hypotheses for that matter. Altogether, I think the framing of the paper needs some work in order to tell a better story of pica in this region with a stronger sense of the study aims and hypotheses. I include more detailed comments below. I think addressing these issues will make for a much stronger paper on a very interesting topic!
Abstract
There is mention that consumption of raw starches is amylophagy, but no mention of geophagy. Either remove amylophagy or add geophagy to be more consistent. Is there a term for charcoal consumption? Also mention that this study includes pregnant women.
Introduction
The hypotheses are laid out in the introduction, but there is no mention that they will be tested. There needs to be a better “storyline” linking the factors that influence pica that are related to the current study, and one or more predictions rather than simply stating that pica is being simply described according to health and sociodemographic factors. Why not also mention that resource scarcity might be playing a role?
Study Population
The authors mention that this study population is food insecure and even collected a measure for food insecurity, but do not link this measure to the hunger hypothesis which has been mentioned in several of Sera Young’s papers. Furthermore, this is particularly important to investigate given that the participants report eating raw starches – many raw starches, such as uncooked rice, have a higher proportion of carbohydrates and protein, which could be an adaptive reason for consuming raw starches when one is food insecure (yet this is called a non-adaptive hypothesis in the discussion which needs to be explained).
Methods
The authors report that consent was given by heads of households. Were the studies conducted in privacy? How are the authors certain that women were not coerced into participating if they did not provide consent themselves?
Are uncooked starches in this region considered food or non-food? How do you know? How would their perception of the extent to which these items are considered food or non-food impact the results? In Placek and Hagen (2013). women were more likely to report consumption of raw starches but not geophagic items perhaps due to local definitions of whether or not these items were foods or non-foods. I think this issue is also discussed in Golden et al. (2012). (These might need to be included as potential references given that they pertain to your study on several different levels).
References:
Placek & Hagen (2013) A Test of Three Hypotheses of Pica and Amylophagy Among Pregnant Women In Tamil Nadu, India
Golden et al. (2012). Pica and Amylophagy Are Common among Malagasy Men, Women and Children.
Analyses
Did the authors control for interviewer effects? Since pica is potentially a taboo behavior in this populations, the authors should test to see if the interviewer and the gender of the interviewers impacted consumption patterns.
There is a hypothesis embedded within the analyses stating that a variable for region was included because the authors predicted regional differences – this should be stated within a hypothesis section and clearly state why they expect this difference. If it has something to do with resource scarcity, then perhaps the authors should run a model including an interaction variable of food insecurity and region to determine differences in pica behavior.
Results
Results indicate that pregnant women were more likely to report engaging in pica. This was a surprise – not because of the finding itself, but because up until this point in the study I did not think pregnant women were being included. Please state earlier in the manuscript that this study includes both pregnant and non-pregnant women. On that note, clearly articulate why women with children less than 2 years old were part of the inclusion criteria.
Discussion
Lines 195-196 “our population was mainly composed of…”
The authors allude to the idea that Region 6 might be more food insecure, yet do not find a main effect for food insecurity. The authors should statistically explore this relationship further.
The discussion about the health correlated of pica are discussed in light of the protection hypothesis. This needs to be brought forth in the introduction as a hypothesis that will be tested.
I also appreciate that the “non-adaptive” hypothesis is discussed in relation to food insecurity. This also needs to be discussed as a hypothesis in the introduction. Furthermore, I would like an explanation as to why eating pica substance is “non-adaptive” in the context of a resource scarcity, especially if starches are able to provide carbohydrates and proteins and clays can sometimes provide bioavailable iron. Please discuss this more.
Be more specific about how culture could impact consumption – maybe discuss how pica is often viewed as a practice for pregnant women, or include an ethnographic example of how pica is tabooed during the postpartum period. I think including more specific details rather just using a “culture blanket” will strengthen this argument.
Limitations
Also note how lumping the starches together could be problematic since they might have different macro- and micro-nutrient contents.
Conclusions
How would future research impact the role of resource scarcity in region 6? Would you use a different measure of resource scarcity? How would you unpack the issue of culture?
Author Response
Dear Reviewer,
We thank you for your thoughtful feedback. Please find a point-by-point response to reviewer comments in the attached Word file.
Thank you,
Esther Chung

Reviewer 2 Report
Thank you for this manuscript. It is novel and interesting and contributes to a poorly researched area of dietary and cultural behavior. Please accept these comments as helpful suggestions. Where applicable I have indicated the line number for your convenience.
Table 1:
The presentation of the data in Table 1 is my primary concern.
Although I agree that the Welch t-test (assuming normality) is a better choice than the Student’s t test given unequal variances and unequal sample sizes, some of your groups are very small. The Welch is not robust with small sample sizes because a normal distribution is suspect. May I suggest for continuous data use the Mann Whitney U test and the Fisher’s exact for categorical data. Although some groups are large enough to use the t-test, for continuity report all with the Mann Whitney or Fisher test.
The statistics reported in the table are not identified. For example identify ‘mean and standard deviation’- see Age, Food Security Score, Household Asset Score, Morbidity Score.
Number of people in the household should be reported by range as there is not a .8 of a person.
The count for geophagy/charcoal/ash is =47 and for amylophagy= 49. The total is =96 yet the count for any pica is 82. Please address the discrepancy.
Figure 1
Figure 1 adds little to your paper and could be removed. Add the information to the text Line 151.
Sample Size
Thank you for the information on sample size calculation. You have described that the calculation was based on fish and child nutritional status not pica. Please report your posthoc power calculation.
Dietary/Pica Measure
The 24 hour recall to assess pica does not include consumption measures (unit) used with the meals. This limits analysis and interpretation. Line 237 should be modified- did you have the power to make this claim? You have no way of knowing how much ash, soil or odowa was being consumed and a 1 day recall will not allow you to infer that the amount the previous day is representative of all days. For some women their intake increases after the rain when the soil has a particular texture and smells appealing.
Also how much would stigma (Line 89) affect the reporting of Nonfoods/Uncooked Foods?
Do you know if participants consider uncooked food (those foods you have categorized under amylophagy) food? This would make them different from soil or ash. Pica is not just consumption but satisfaction of a craving. Not necessarily a neurological or sensory problem (Line 58)
Was odowa purchased or self supplied? Is it available in all regions?
Methods Section
Methods: Less on the study site and fish history and more on the availability of substances women might consume from the regions and what they do during the day. Do men and children also practice pica in the regions? This may influence women’s patterns. I am also not convinced that breastfeeding moderates pica- women are in a different physiological state during breastfeeding- different from pregnancy but they are also performing different tasks. Perhaps the business of caring for a baby including a lot of time breastfeeding, and washing limits engaging in pica.
I believe the lack of expected relationships (Line 288) while ambitiously anticipated, is a function of divided samples, and a weak measure of pica. Human behavior is complex.
Author Response

(The authors gave the same response as above.)

Round 2
Reviewer 1 Report
Thank you for taking time to address all of my comments. Great job. I look forward to seeing this in print.
Author Response
Dear Reviewers,
Please find our responses in the attached Word file.
Thank you,
Esther Chung

Reviewer 2 Report
Thank you for your responses to my earlier review. May I suggest the following: Line 55 and 64 add references as is the first mention.
You mention that data collection occurred from March- December. Do you have any reason to think that January to Feb would be different? A seasonal effect perhaps- rainy vs dry season?
Figure 1 Please include how many women consumed two or more substances after the total number 81.
Line 192-195 Need references and the measures (food record, repeated dietary recall) used in the other studies to support your claim. What is 'longer'.
Table 1 In the first review I did not comment on p values. I Strongly suggest p values remain.
Line 273 Do you mean "longer" rather than shorter.
Line 291 Remove 'is critical" as you have not indicated what is critical.
In your analysis where you have small sample sizes please indicate what effect this may have on the result- over estimate? under estimate? When the confidence interval is wide there is a lack of confidence in the result. What does that mean for the result you are reporting that has a wide confidence interval.
"odds of amylophagy were higher among currently pregnant women (OR=4.31, 95% CI: 1.24-14.96)."
I understand the challenge of a study within a larger study. You are restricted to the latter's timeline and imperative. The one day dietary recall about the substances of pica did not include frequency, an estimate of amounts nor the impetus to engage in the behaviour. I find you have stretched the data as much as possible.
You report that food insecurity was common so hunger could be the reason pregnant women who have increased appetites are eating non food items. A similar reality to that during WW2 in Europe when people in embargoed countries consumed many non food items eg. mixing yellow clay with water to make 'butter'.
I appreciate your work in revising your paper and the patience you have shown with my comments.
Author Response

(The authors gave the same response as above.)

Round 3
Reviewer 2 Report
Response 5: Thank you for your comments and references regarding the inclusion of p-values on Tables. I will leave this to the editor to decide.
Response 9: Line 211 does not address this. When a 24 dietary recall is properly done it includes detailed information on food and drink consumed on a given day and the total amount of each food and beverage. It can be argued that pica is not dietary intake and that the 24 hour dietary recall is not valid for non-food items. It is known that pica behaviour increases during pregnancy sometimes to 'dry up' salivary secretions, to 'colour' the baby or to ensure the baby has beautiful hair. These and the limitations of the 24 hour dietary recall as a representative measurement of intake should be mentioned.
The science of human dietary intake has been plagued by the lack of reliability and validity and accuracy in its measures.
Author Response
Dear Reviewer,
Please find our response to your comments in the attached Word file.
Thank you,
Esther Chung
